# Supporting Cross-Company Networks in Workplace Health Promotion through Social Network Analysis—Description of the Methodological Approach and First Results from a Model Project on Physical Activity Promotion in Germany

**DOI:** 10.3390/ijerph18136874

**Published:** 2021-06-26

**Authors:** Andrea Schaller, Gabriele Fohr, Carina Hoffmann, Gerrit Stassen, Bert Droste-Franke

**Affiliations:** 1Working Group Physical Activity-Related Prevention Research, Institute of Movement Therapy and Movement-Oriented Prevention and Rehabilitation, German Sport University Cologne, Am Sportpark Muengersdorf 6, 50933 Cologne, Germany; c.hoffmann@dshs-koeln.de (C.H.); g.stassen@dshs-koeln.de (G.S.); 2IQIB–Institut für Qualifizierende Innovationsforschung & -beratung, Wilhelmstraße 56, 53474 Bad Neuenahr-Ahrweiler, Germany; gabriele.fohr@iqib.de (G.F.); bert.droste-franke@iqib.de (B.D.-F.); 3Institute for Occupational Health Promotion, Neumarkt 35-37, 50667 Cologne, Germany

**Keywords:** workplace, health promotion, physical activity, organizational network, social network analysis, two-mode network

## Abstract

Cross-company networking and counseling is considered to be a promising approach for workplace health promotion in small and medium-sized enterprises. However, a systematic and empirical approach on how such networks can be developed is lacking. The aims of the present paper are to describe the approach of a social network analysis supporting the development of a cross-company network promoting physical activity and to present first results. In the process of developing the methodological approach, a common understanding of the nodes and edges within the project was elaborated. Based on the BIG-model as the theoretical framework of the project, five measuring points and an application-oriented data collection table were determined. Using Gephi, network size, degree, and distance measures, as well as density and clustering measures, were calculated and visualized in the course of the time. First results showed a continuous expansion and densification of the network. The application experience showed that the application of social network analysis in practical cross-company network development is promising but currently still very resource intensive. In order to address the current major challenges and enable routine application, the development of an application-oriented and feasible tool could make an essential contribution.

## 1. Introduction

Since the Prevention Act from July 2015 [1], workplace health promotion (WHP) has been a crucial setting for prevention in Germany as it is considered to a promising approach to reach relevant target groups, for example men, workers with health problems and/or older workers, people with multimorbidity, or unskilled workers [2,3,4]. Thereby, physical activity is the largest field of action, comprising about 70% of behavior-related and 50% of environmental measures [2]. The growing relevance of WHP is also reflected in a significant increase in expenditure by the statutory health insurance funds (SHI) as the expenditure in this field tripled from 2015 (76 million euros [5]) to 2019 (240 million euros [2]).

Nevertheless, the number of employees reached (2015: 1.20 million; 2020: 2.28 million) [2,5] continues to fall well short of expectations. At present, it must be assumed that only about 5% of the approximately 33 million employees in Germany who are subject to statutory health insurance [6] are reached by WHP financed by SHI. Therefore, various explanations are discussed. On the one hand, the low utilization of WHP could be attributed to influencing factors on an individual level such as a lack of motivation, low health literacy, low self-efficacy, and lack of health awareness [7,8,9]. On the other hand, this could also be due to organizational or structural reasons. Even though the number of companies reached with SHI prevention programs has more than doubled from 2015 (11,000 [5]) to 2019 (23,000 [2]), only a fraction of the approximately 7.8 million companies in Germany [10] are reached with WHP financed by SHI. This could be due, amongst other things, to the fact that in Germany more than 95% of all companies are small and medium-sized enterprises (SME) [10]. Despite numerous efforts, SME are still a relatively underrepresented area of SHI-financed WHP. Reasons include lack of time and/or human resources as well as the incurrence of additional costs by WHP [11,12,13]. In addition, lack of program know-how of the executives, low interest of the employees, and an uncertain return on investment are identified as reasons for underrepresentation of SME [14]. The fact that SME often lack the resources for WHP [11,13] appears to be a solid argument for the implementation of cross-company networks to bundle competencies and pool resources.

In addition to the established WHP offers of the SHI from the areas of *advice on health-promoting work design* and *health-promoting work and lifestyle,* a third WHP field of action was developed, which addresses *cross-company networking and counseling* to support SME in particular in promoting WHP [15,16]. Thereby, cross-company networks are defined as organizational associations of different companies that jointly offer workplace health promotion measures and are supported by external providers. Taking into account the specific challenges of SME for WHP, cross-company networks aim at providing a low-threshold access by supporting companies that do not have enough resources for WHP through networking. In order to achieve this, the SME network can also be supported by local stakeholders such as representatives of social insurances and local health care providers [15,17]. Even though WHP remains to be a challenge for SME, the cross-company network approach seems to be promising as increasing numbers of SMEs utilize SHI-funded WHP. Since the implementation of the WHP-offer *cross-company networking and counseling* in the year 2014, the share of SME in WHP financed by SHI increased from 21% [18] in 2013 to 27% in the year 2019 [2].

Despite the strong assumption that cross-company networking and counseling might be a promising approach for WHP in SME, up to now, there is no empirical data on how such networks can be systematically developed. Therefore, the theoretical concept of social network analysis (SNA) offers a promising framework [19,20,21] and strategy to investigate social structures [22]. Basically, networks are described as systems in which different elements or nodes are connected by edges [23]. SNA enables the analysis and visualization of network structures in terms of nodes, respectively, actors (e.g., natural persons or legal entities, such as companies, schools, etc.) and edges (syn.: ties or links), representing the relationships or interactions that connect them [24,25]. With this, SNA aims, for example, at conceptualizing social structure as a network by connecting members and channeling resources with edges as well as focusing on the characteristics of edges rather than the characteristics of the individual actors (nodes) [26]. According to Freemann [27], four features are generally used to describe the network and explain the relationships within the network: structural intuition, systematic relational data, graphic images, and mathematical or computational models [27]. In this way, SNA enables the visualization and analysis of the meso-level relationships in which the nodes are embedded.

Despite the growing application of SNA in various fields, e.g., decision making processes in commercial organizations [28] and various research fields [29], the application of SNA in the fields of sports science [30] and health promotion or health care [31,32,33] has been rare so far. Therefore it is the aim of the present paper (a) to describe the SNA approach aiming at supporting the development of a cross-company network promoting physical activity in a German model project and (b) to present the first results.

## 2. Materials and Methods

### 2.1. The KomRueBer Project

The KomRueBer project is part of the funding priority “Exercise and the Promotion of Physical Activity” by the Federal Ministry of Health (BMG) [34] and started in July 2019 (German Clinical Trials Register (DRKS)-ID: DRKS00020956; 18 June 2020).

Briefly summarized, the project deals with the development, implementation, and evaluation of a cross-company network for promoting physical activity. Setting of the project is a technology park in Germany with around 90 different-sized companies. A positive approval was received from the ethics committee of the German Sport University Cologne (reference numbers 120/2019 and 068/2020). Overall, the KomRueBer research project comprises three stages over a total period of 36 months: (1) the conception stage, (2) the implementation stage, and (3) the evaluation stage. The procedure and the results of the development of the cross-company network as well as the multicomponent intervention during the conception stage were already published in Hofmann et al. 2020 [17]. The implementation stage and the accompanying evaluation stage of the project have been ongoing since March 2020 until May 2022.

The working procedure during the conception stage as well as the implementation stage was based on the theoretical framework of the BIG manual [35]. BIG is the German-language acronym for “Movement as an Investment for Health” and has been a model project for the promotion of physical activity among socially disadvantaged women [36]. In general, the BIG manual describes the procedure for the participatory development of physical activity promotion projects. A core element of BIG is the participation of the target group and relevant stakeholders. Besides, a cooperative planning process in order to develop corresponding health-promoting measures is of particular importance. The procedure of BIG is divided into five phases: (A) finding, (B) preparation, (C) cooperative planning process, (D) intervention process, and (E) ensuring sustainability [35]. Phase (A) finding is about creating the prerequisites for the realization of the project and arousing the interest of potential partners. During phase (B) preparation, all important measures for a successful implementation—like the management of responsibilities, target definition, or contacting the target group—are taken. The cooperative planning (phase (C)) comprises the participatory development of measures and the preparation for implementation. Finally, the implementation of the measures (phase (D)) is controlled and the sustainability of the project (phase (E)) is ensured (see [35]).

Additionally to the BIG manual, a so-called manager of the cross-company network, who is employed by an external WHP provider, plays a key role in the KomRueBer project. The manager of the cross-company network is responsible for the realization of the different work steps within the BIG manual and bears responsibility for the coordination of the network within the implementation phase. In detail, this includes the coordination and moderation of the individual work steps and phases described in the BIG manual. The manager acquired the various stakeholders for the project, leads the different events, and plans as well as coordinates the measures and activities with the various stakeholders with whom he is in close contact. The first acquisition of stakeholders was conducted as part of the application process for the research project and, thus, before the start of the KomRueBer project. Thereby, the manager of the cross-company network was not involved.

### 2.2. Objective and Research Questions of the SNA within the KomRueBer Project

The aim of the SNA within the project is to display the conception phase (A–C) as well as the implementation phase (D, E) of the cross-company network promoting physical activity. In order to ensure the sustainability and transferability, this methodological component aims at providing the foundation for the development of an application-oriented SNA tool to support a manager of the cross-company network in developing and implementing the network for the promotion of physical activity within the framework of the model project. The related questions for the present paper are as follows:How can SNA be applied to depict the development and implementation of a cross-company network for promoting physical activity?How has the cross-company network for the promotion of physical activity developed over time?

### 2.3. SNA Approach

We assumed that the relationships (edges) within our network are reciprocated, and consequently, we decided to conduct global non-directional network analysis to consider all relations between the actors in the network [22].

#### 2.3.1. Defining Nodes and Edges

In the first step, we elaborated a project-related understanding of the nodes and edges. The nodes (actors) at the analysis level represent the participating organizations of the cross-company network promoting physical activity. Every organization attending a network event (see Section 2.4.1, Table 1), one time or repeatedly, is counted as a network member. We defined the participating organizations as the primary node type, in accordance with the focus on the workplace in KomRüBer.

#### 2.3.2. Identifying and Defining Network Members

The first network members from the field of physical activity and health promotion as well as companies were already project partners at the beginning of the project as they already supported the application process. Further stakeholders were identified by the participating partners during a participatory workshop in the finding phase (A) (see [17]). All stakeholders considered in the KomRueBer project were divided into five groups: *company representatives, exercise providers and network partners from public, society/politics, and economy. Company representatives* were defined as representatives from a company located in the technology park, for example general managers, head of human resources, or members of the work council. They actively represent their company in the project by participating in workshops, working groups, or common activities. *Exercise providers* are, for instance, fitness studios, physiotherapists, or clubs that offer various physical activity promotion measures for the project. These measures can be implemented both on site and digitally. Thereby, a distinction can be made between regional and supra-regional exercise providers with a main focus on regional providers. *Network partners from society/politics* were defined as actors who are relevant for the development and implementation of the network and who belong to the fields of society and politics. These can include, among others, health insurance funds, the pension fund, or municipal administration. Likewise, *network partners from public or economy* were defined as relevant actors for the project from the respective fields. The latter can include, for example, business development agencies, town and country planners, or sports article retailers, whereby network partners from the public can be media representatives or other public persons.

As a methodological tool, events (e.g., meetings and workshops) were defined as a second node type during the course of the project (see Section 2.4.1; Table 1). The edges in our SNA represent the participation of the corresponding actors in networking events.

With this SNA approach described, we monitored the conception stage of the KomRueBer project and, to date, the first period of the implementation stage. Subsequently, the procedure for data collection and data processing is described (Section 2.4.1).

### 2.4. Data Collection, Preparation, and Processing

#### 2.4.1. Data Collection

Data for the present evaluation were collected in the period from August 2018 to October 2020. In order to collect the data, five points in time as respective work steps were defined within the framework of the BIG model (see Table 1). This procedure allows us to compare subsequent points in time with earlier points in time.

By means of the Letter of Intent (T1), different stakeholders on site expressed their interest in the participation of the KomRueBer project. The Letters of Intent were obtained during the application process of the research project and thus before the beginning of the project. This was obligatory in the context of the application and enabled us to have an appropriate basis for the start of the project. Work step T2 comprises the on-site information and consultation of further project partners by the manager of the cross-company network. Therefore, the network manager visited various companies, potential network partners, and exercise providers on site. The network manager elucidated the project, answered questions concerning the project, and enquired about the interest in participation. Potential partners were identified through investigation or were recommended by already participating partners. The Stakeholder Workshop July 2019 (T3) lasted two hours and was realized with different stakeholders on site. The aim of the workshop was to collect initial ideas for physical activity promotion measures for the cross-company network. Within the second stakeholder workshop (T4), which lasted four hours, the stakeholders pre-finalized the multicomponent intervention for promoting physical activity. The methodology of the workshops has already been described by Hoffmann et al. [17]. In October 2020 a first steering group meeting with the stakeholders was realized. On the one hand, this two-hour meeting included the discussion of the previous measures. On the other hand, potentials for improvement concerning the implementation of measures were discussed and new activities for the cross-company network were presented. All work steps from T2 to T5 were organized and carried out by the manager of the cross-company network.

#### 2.4.2. Preparation of the Data

In order to collect the raw data in a practicable way during the course of the project, the manager of the cross-company network maintained a personal list of the participants in the individual work steps (see Table 2). This list was prepared on the basis of the memorandums of the conversations (T2) as well as the lists of participants (T3–T5). In step T1, the information on the letter of intents was used to collect the relevant data. This Excel sheet contained the following information: person name, affiliation, stakeholder group, and participation at a work step (“1” = participation, “0” = non-participation). This approach allows for the collection of network data at the time a process step is conducted.

#### 2.4.3. Processing of the Data

This procedure for data collection and preparation made it possible to display a two-mode network with two different sets of nodes (organizations, events; see Section 2.3). In the next step, the data collection table was anonymized and aggregated as the basis for the SNA data analysis. The following data format was used: organization, stakeholder group, and participation at the following events (yes = 1; no = 0): T1: Letter of Intent; T2: On-site information and consultation; T3: Stakeholder workshop; T4: Stakeholder workshop II; T5: Steering group (digital meeting).

To evaluate the development of the cross-company network for the promotion of physical activity over time by SNA methods, a two-mode network of events and participating actors was applied. Afterwards, we projected the two-mode network to a one-mode network to be able to calculate certain network-specific measures. The projection of the data on a one-mode network allows for the calculation of network measures—for individual nodes and for the overall network. This was a necessary step to be able to calculate comparable and unbiased metrics for the nodes in focus, i.e., the organizations. For this, the event nodes were projected to edges that connect organization nodes. This results in a one-mode network of organizations that are connected by their shared participation in a network event (work step). While the organizations of the two-mode network were connected to the events, they are in one mode—after the projection—connected to each other on the basis of having something in common: participating in a network activity or attending an event. The visualization as well as the calculation of different quantitative network measures based on the aggregated data is presented in Section 2.4.

### 2.5. Analysis

The visualization and calculation of the metrics was done using the free software Gephi (https://gephi.org/users/download/, downloaded on 11 March 2019). As the aggregated data set (see Section 2.4.2) was already properly formatted, the node and edge lists could be imported to Gephi. Metrics were calculated in Gephi 0.9.2. Global network measures were calculated for the present evaluation. The selected measures calculated are listed in Table 3.

## 3. First Results

The first results presented below comprise the period from the beginning of the project (T1), the conception phase (T2, T3, T4), and the start of the implementation phase (T5). At the beginning (T1), a total of 9 actors were involved (4 exercise providers and 5 network partners); the number of actors involved increased to 23 during the course of the project so far (T5), of which 7 were companies, 8 were exercise providers, and 8 were network partners. The following chapters show the visualization (Section 3.1) and the network metrics (Section 3.2) in the course of the project so far.

### 3.1. Visualization

Figure 1 shows the development of the cross-company network for the promotion of physical activity over time. The two-mode networks (top row) display the participation of the actors (nodes) at an event and the one-mode networks (projected two-mode; bottom row) display how the organizations are connected by their common participation at events.

The graphics for the two-mode network showed an enlargement of the network, measured by the number of participants (nodes) and connections (edges) over time. In addition to the increase in the total number of connections in the network, there was also an increase in the average number of connections per participant. At time step T5, the network was well connected as only a few organizations (10 out of 23) only participated once. Overall, 13 organizations participated two or more times: 5 organizations participated two times, 5 three times, and 3 participated four times.

The one-mode networks showed that a denser network of about 7, 8 to 12 organizations was formed (bottom row, T3–T5). While at T5, the network size was 23 organizations, 12 of them were connected more densely and formed a core network of companies and organizations that have met each other two or more times at network events. It is noticeable that at the events T2 and T3 only one and two organizations have participated twice, while at the events T4 and T5, more organizations met more often.

### 3.2. Network Metrics

The projection of the events onto the edges (one-mode) allowed for the calculation of specific network metrics, which are shown in Table 4. The metric measures confirmed the visualized network expansion and densification. Just as seen in the visualizations, the number of nodes and edges increased over time and there was also an increase in the average number of connections per participant and a growing weight of connections. Overall, the network metrics showed an increase of the network from the initial 9 actors to 23 actors (number of nodes) and an increase in the number of connections (number of edges) between the actors from 36 to 153 in the same period.

Regarding the degree measures of the network, the average degree, which describes the average number of connections of a single actor to others, increased from 8 (T1) to 13 at T4 (Stakeholder workshop II), but decreased slightly between T4 and T5, which was between the Stakeholder workshop II and the digital meeting of the steering group. Although the average degree did not increase from T4 to T5, the average weighted degree increased continuously, indicating that the edges of organizations that met more frequently were strengthening.

The distance measure (network diameter and average path length) underlined the closeness of actors in the network. The maximum distance between two nodes in the graph (network diameter) was two and the average path length, describing the average distance between all pairs of nodes varied between 1.4 and 1.5.

Network density describes how many edges between actors exist compared to how many edges between actors could maximally exist. At T1, the network was complete in the sense that all actors met for the first time, and therefore, every actor was connected to every other actor (network density = 1). Since not all actors from T1 participated at T2, the value decreased (0.529) then increased over the course (T3: 0.558; T4: 0.641), and decreased again slightly towards the last time step shown here at T5 (0.605). This implies that around 50%–60% of the possible connections in the network developed during the project. The density and clustering measures showed a continuous increase in the total number of triangles (84, 168, 305, 528, and 538) even though the increase was weakening. In contrast, the clustering coefficient decreased from 0.969 (T2) to 0.848 (T5), which reflects a continuous enlargement.

## 4. Discussion

### 4.1. Summary of Findings

Our paper describes the methodological approach of an application-oriented SNA in the context of cross-company networks for health promotion and the first results on the structural development within a model project on physical activity promotion. The results available so far in terms of the development of the cross-company network for the promotion of physical activity over time are promising. Overall, it can be seen in the figures that the network grew from two weakly connected networks at T2 to a strongly connected network with a clearly established core. The development visible in the figures was confirmed by the metrics used. The measures proved an increase of strength in the network connections while the network was continuously growing. Our results indicate a continuous expansion and densification of the network could be realized while preventing the building of sub-clusters.

### 4.2. Classification of the SNA Approach in the State of Research

Despite the known potential of networks within the health sector and the growing importance of multisector collaboratives for improving population health [19,39], the implementation of SNA in the context of healthcare [28] and the application of a network approach for developing and implementing health interventions [40] is still considered rare. Existing SNA research in healthcare mainly focuses on individual health behavior and relationships [41], disease transmission [40] and socially determined health inequalities [42]. In recent years, there has been an increasing focus on SNA research at the organizational level [19,28,43,44,45], which is where the KomRueBer project can be classified. Our network can be understood as an inter-organizational network as it focuses structures between organizations (e.g., collaboration between exercise providers, companies, and network partners), but also comprises the affiliation network perspective by assessing the participation of actors at events [30]. As usual with affiliation networks, we combined the approach of a two-mode network and a one-mode projection to analyze network metrics [30].

### 4.3. KomRueBer Network Development

Regarding the network development from an SNA perspective, the comparison of the figures on the project progress (see Section 3.1, Figure 1) is interesting: The network, which was initially created by the LOIs (T1), was expanded by additional participants in the second step (T2). There only was one participant who was present at both events. Thus, the networks at T1 and at T2 were almost separate from each other. With T3, two things happened: the network from T1 was strengthened and the upper network became better connected (lower part of the third figure in the bottom row). At T4, the connectivity was further strengthened and a small core also developed in the upper part. At T5, this core was more strongly connected to the core of the lower network, so that the cores were well connected. This showed that the network was strengthened as a whole. Overall, the figures show that the network grew from two weakly connected networks at T2 to a strongly connected network with a clearly established core. The figures show step-by-step the densification of connections between the two sub-networks, the subsequent development of a core in one network, and finally the connection of two cores to one larger core, indicating increasing stability of the network with a mixture of strong and weak ties.

The metrics underline the visual impression: The number of participating organizations (nodes) was continuously increasing, but at the same time, the metrics showed an increasing “average weighted degree”, indicating that the network was getting stronger. Even though the increase of the network size is basically due to the fact that the events were added up, a growing number of organizations participated in network events over time, which can be seen in the non-linear increase in edges between T1 and T4.

Additionally, the network density metrics and the increasing number of triads confirm that the events T2, T3, and T4 connected actors more densely by every step. The number of participating organizations per single event increased from T1 to T4 but decreased at T5 (digital meeting in October 2020), most probably due to the COVID-19 pandemic. The slight reduction in network density might mainly be due to the growing number of participants.

Nevertheless, the network density also points to networking potential within the network, as only up to a maximum of 64% (T4) of the possible connections in the network have been utilized in the course so far. However, it must be taken into account that in SNA, the optimal point of density is not 1, but below. Too strong an embedding of the stakeholders (over-embeddedness) might impede the performance of a network. For example, it can be beneficial in some phases of the KomRueBer network development to have new participants every now and then in order to get new impulses for the network, even if it temporarily reduces the network density. This also applies if these participants are only weakly connected, e.g., are only at the event once (“strength of weak ties”). It seems as if the network consolidates at a mid-range of density, which may be a good level to avoid over-embeddedness and still allow it to have weak ties, which may bring new impulses from the outside. The optimal density point, however, must be determined by the network manager in terms of content. SNA data cannot give any recommendations on this yet on the basis of our first model project.

The distance measure (network diameter and average path length) underlined the closeness of actors in the network, which builds a good basis for the flow of information and building of trust. The diameter of the network was always at a very good level with a maximum of 2. The interpretation of the low values of the distance measures (network diameter and average path length) suggests a mostly direct connection between the organizations. This is rather positive in the context of the planned network for promoting physical activity. As the network relies on communication, the sharing of resources and social capital, the short distances between the actors might ease the flow of information and support the building of trust.

Regarding the degree measures, so-called “bridge organizations”, which are defined as organizations that have participated in several events, seem to be of particular importance. At T4, 13 of the 22 organizations had already participated in two or more events. In terms of network promotion and flow of information, these organizations joined the information and contacts from several events. Consequently, when developing networks, the network manager should aim at establishing a maybe small but stable core of participating organizations that take part in multiple events. Looking at the average degree shows that the conceptual objective of an event must always be taken into account for its evaluation. The fact that this value did not increase further at T5 shows that the network was not extended further. This corresponds to the objective of event T5 (steering group digital meeting), which was not to expand but to stabilize the existing network. The fact that the objective of the T5 event could be achieved is reflected in the increase of the average weighted degree that takes edge weight into account: at T5, particularly, the edges of organizations that met more often both strengthened and stabilized the existing network.

Overall, these developments reflect that the network was steadily enlarged over time, while the numbers of new connections declined in the last meetings. This indicates that the network development enters a new phase from building up network connections to strengthening the network and communicating its work. From T5 onwards, it is therefore important from a network perspective to create a good balance between further strengthening the network and recruiting new participants, and thus, also impulses for further development. Again, this decision depends on practical aspects of content rather than SNA metrics. Guiding questions for an appropriate decision might be as follows: Am I satisfied with the number and type of physical activity interventions offered? Do I need additional actors or new ideas? Do I have clear responsibilities and do I continue to involve the network participants who are already committed?

### 4.4. Practical Implications for the Development of Cross-Company Networks in WHP

The practical relevance of the results lies in the possibility of monitoring and evaluating the network development. Especially the visualization might be a promising tool for intensifying the cooperation of organizations, which can be drawn upon when it comes to sustainability issues as well as transferring the idea of a cross-company network promoting physical activity to other regions.

To enable the routine application of an SNA in networks for cross-company physical activity and health promotion, one focus of our work was on intensive cooperation between WHP practitioners and network analysts. Based on the assumption that SNA might be a promising reflective tool for public health practitioners [39], we anticipate that SNA could be used by WHP practitioners to examine their own practice. Our experience during the project confirmed that the application of SNA was useful for the manager of the cross-company network to realize that the network can be developed with a strategic thought in order to strengthen the network composition and the collaboration within the network [39]. Thereby, major challenges in the cooperation between SNA science and practice [46] became apparent, and this turned out to be a very resource-intensive process. Considerable challenges became apparent, for example, in the development of an understanding of the goal of the cross-company network on the part of the network analysts and, on the other hand, in the understanding of the content of the network measures for the practical work as a network manager. It became obvious that knowledge of the respective different content-related goals and structural framework conditions is indispensable. WHP practitioners, and therefore network non-specialists, must be very confident in their ability to learn about the network concepts and methods in regard to achieving a productive effect on their application-oriented questions [29]. Given the variety of methodological and scientific options of SNA, however, it is important to consider first and foremost the feasibility of implementation (e.g., data collection) and the preparation of results for practitioners in terms of content.

With our work in the KomRueBer project, we hope to solve these challenges in cooperation and to develop network-analytical methodological and also content-related foundations from the field of WHP for the resource-efficient implementation of an application-oriented SNA in the context of cross-company health promotion for subsequent projects. In this context, the SNA results on network scale-up and development were consistent with the practical experience of the manager of the cross-company network. Against the background that networks in physical activity and health promotion do not usually develop on their own, the present results also reflect the enormous workload of the manager of the cross-company network during the network development phases. This in turn also underlines the need for a model-based approach to building a network. Our orientation to the BIG model [35] thereby proved to be helpful, as it already suggests several relevant stakeholders for network development in physical activity promotion. From our experience, the accompanying SNA in KomRueBer could, as a first step, bring to the attention of those responsible decision-makers in the fields of prevention and WHP that active network management, implemented in our project through the role of the so-called manager of the cross-company network, is a central success factor for the establishment of a cross-company network for the promotion of physical activity and health.

### 4.5. Study Limitations and Strengths

Our work in this area currently has some limitations. Basically, this article is based on the SNA expertise of the author team with regard to the technical realization and interpretation of social network analyses. We cannot provide a social science reference to the planning and execution of network interventions. Furthermore, our analysis was limited to contacts during the defined events. Furthermore, the individual actors were not additionally asked about contacts with each other between the events. Second, on the basis of our first results, we cannot yet make any recommendations on the optimal composition of the actors in a cross-company network. This will be part of the analyses at the end of the project, considering the different stakeholder groups. We have also not yet integrated organizational sizes and types into our analyses. For further research work in this area, this could be interesting in order to understand if and how the composition of the network in terms of organizational types changes over time. In further projects, this could be promising to assess, if the network is well composed and therefore a stable network of various organizations.

Our approach also does not allow for any conclusions to be drawn about the relationship between physical activity measures offered (e.g., the acceptance, utilization, or its effectiveness) and the course of the network development. We assume that there is no guarantee that time spent on network development will necessarily improve the physical activity behavior of the target group. However, as various different functions and types of social networks might be relevant related to the respective health outcomes [45], a solid cross-company network for promoting physical activity might be an essential environmental-oriented condition for promoting physical activity in employees. However, future SNA-research should seek to go beyond the merely descriptive to implement and evaluate SNA-based interventions. In the long run, the association between network structure and the effectiveness of interventions promoting physical activity, as well as the effectiveness evaluation of the use of the network tool in network development, would then be promising in subsequent projects.

The strength of our model project lies, to our knowledge, in the first targeted application of SNA in the context of health and physical activity promotion in cross-company networks. Thereby, our study aimed at systematically understanding and describing the development of cross-company networks promoting health and physical activity in the course of time. From a methodological perspective, we were able to make first experiences in exploring the feasibility and utility of SNA in the practice of cross-company health promotion. Thus, our project makes a contribution to the research gap of the systematic application of SNAs in designing, disseminating, and implementing health interventions [33]. A further innovative contribution of our study is the contribution to the technical development of an application-oriented SNA tool. Therefore, we also described an approach for data collection and extracting relevant information from social network data to support the understanding of network development in WHP.

## 5. Conclusions

Overall, our model project highlighted, amongst other things, the importance of active network management and the relevance of defining appropriate competencies for this role. Thereby, our project experiences led us to the suggestion that SNA could be helpful for WHP practitioners to systematically build cross-company networks in health promotion and physical activity promotion. With our work in the KomRueBer project, we have so far been able to contribute to identifying underlying challenges in the process of implementing SNA in cross-company WHP, and to contribute to the systematization and quality assurance of this important prevention approach. In order to make a further contribution relevant to practice, we are currently working on an application-oriented SNA tool that can be used by practitioners to accompany the development of cross-company networks promoting physical activity. In doing so, we aim to create a foundation for WHP practitioners to become familiar with the potentials, theories, and key metrics for applying SNA in cross-company networks for health promotion.

## Figures and Tables

**Figure 1 ijerph-18-06874-f001:**
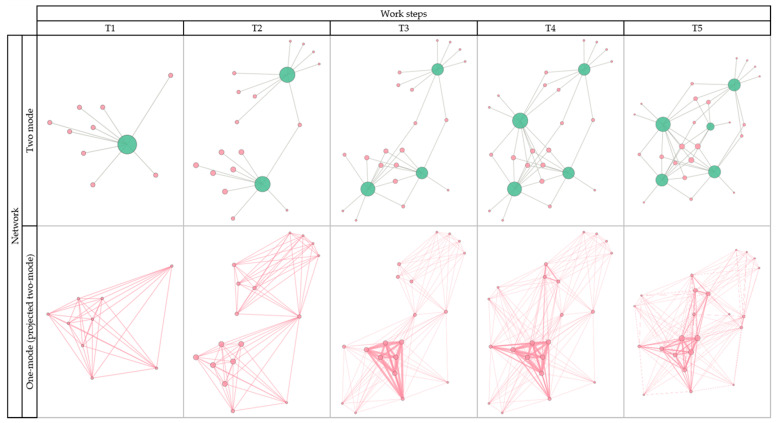
Visualization of the network at the five work steps: T1 (letter of intent), T2 (on-site information and consultation), T3 (stakeholder workshop), T4 (stakeholder workshop II), and T5 (steering group (digital meeting)) (red = organizations; green = events); top row: two-mode network; bottom row: one-mode (projected two-mode) network.

**Table 1 ijerph-18-06874-t001:** Work steps and points in time of the data collection.

Phase According to the BIG Model	Step	Description of the Event	Date/Period
Pre-conception phase (proposal)	T1	Letter of Intent	August 2018 (before project start)
Conception phase	T2	On-site information and consultation	July–September 2019
	T3	Stakeholder workshop	July 2019
	T4	Stakeholder workshop II	January 2020
Implementation phase	T5	Steering group (digital meeting)	October 2020

**Table 2 ijerph-18-06874-t002:** Data collection table (exemplary).

Name of the Person	Affiliation	Stakeholder Group	Participation in the Work Step				
			T1	T2	T3	T4	T5
AA	aa	Company	1	1	0	0	0
BB	bb	Exercise provider	0	1	0	1	0

**Table 3 ijerph-18-06874-t003:** Calculated global network measures (see [37,38]).

Network Measure	Description
**Network size**	
Number of nodes	Total number of nodes
Number of edges (ties)	Total number of edges (ties)
**Degree measures**	
Average degree	Average of individual values of degree centrality (number of direct connections).
Average weighted degree	Average of individual values of degree centrality, considering edge weight (edge weight: sum of the edges between two nodes).
**Distance measures**	
Network diameter	The maximum distance or maximum number of steps between any pair of nodes in the graph (longest path).
Average path length	Average number of steps along the shortest paths for all possible pairs of network nodes.
**Density and clustering measures**	
Network density	Number of edges divided by the total possible edges; a measure of well connectedness of a network (complete = 1).
Total number of triads	A triad is given if a node’s two neighbors are connected to each other.
Average clustering coefficient	Degree to which the nodes of a network tend to cluster together; calculated by using “number of triangles” (complete = 1).

**Table 4 ijerph-18-06874-t004:** Changes of network measures over time (one mode).

	T1 Letter of Intent	T1 to T2 On-Site Information and Consultation	T1 to T3 Stakeholder Workshop	T1 to T4 Stakeholder Workshop II	T1 to T5 Steering Group (Digital Meeting)
Number of nodes	9	17	20	22	23
Number of edges	36	72	106	148	153
Average degree	8.0	8.5	10.6	13.5	13.3
Average weighted degree	8.0	8.5	12.7	17.5	18.1
Network diameter	1	2	2	2	2
Average path length	1.000	1.400	1.442	1.359	1.395
Network density	1.000	0.529	0.558	0.641	0.605
Average clustering coefficient	1.000	0.969	0.915	0.854	0.848
Number of triads	84	168	305	528	538

## Data Availability

The data presented in this study are available on request from the corresponding author.

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
