# Peer review of "Supporting Cross-Company Networks in Workplace Health Promotion through Social Network Analysis—Description of the Methodological Approach and First Results from a Model Project on Physical Activity Promotion in Germany"

_ijerph, 2021, doi:10.3390/ijerph18136874_

Round 1

Reviewer 1 Report

Thank you for the opportunity to review the paper “Supporting Cross-Company Networks in Workplace Health Promotion through Social Network Analysis – Description of the Methodological Approach and first Results from a Model Project on Physical Activity Promotion in Germany.” It describes an interesting application of social network analysis (SNA) in the context of cross-company initiative to support workplace health promotion in small to medium enterprises.

The authors describe their process of analysing a bipartite network of inter-organisational relations, as captured by attendance at five separate curated networking events. They proceed with a descriptive analysis of the network and a critical evaluation of the utility of their approach. The paper illustrates a learning process about the implementation of an SNA in this context and points to interesting applications in future practice, specifically the strategic planning and management of organisational networking interventions.

The strength of this paper lies in this final, critical evaluation of its data collection process and its limitations. I agree with the authors that what they observed was merely limited to representatives’ attendance at the defined events without any further analysis of interpersonal relations or any substantive measures of effects on participants’ practice. As the authors point out considerable challenges became apparent in deriving meaningful insights from network measures for both the network analysts and the network manager.

On this basis, I believe the paper has merit, but requires substantial revision. It offers a comprehensive list of descriptive, longitudinal statistics, but fails to point out their relevance for this particular networking project, given the nature of the relationships they capture. Consequently, it offers little new insight as empirical evaluation of inter-organisational activities. However, it does capture the deliberate curation process of a collaborative, inter-organisational network and with a review of available literature on the planning and execution of network interventions, this paper could make a very interesting contribution.

Reviewer 2 Report

The paper addresses a less researched topic of corporate network development in the context of workplace health promotion, which is relevant to public health and to the scope and interest of this journal. However, many areas of the introduction, theoretical framework, terminology used, methodology, and results/discussion require clarification. With extensive revision, this paper could be impactful.

Below are my recommendations for major revisions:

  • Define the term “cross-company networks related to workplace health promotion” at the outset (before stating the research questions). The term is misleading if it is not clearly defined at the outset, because the network refers to organizational attendance of information sharing events rather than participation in group health promotion/physical activity sessions. Also, it needs to be clarified that this is a network of organizations (companies) and not employees within/across these companies.
  • Clarify the sentence on Page 2 Line 70. The meaning is unclear at the moment: “The development of the cross-company network approach in WHP could be one reason why, after all, meanwhile 27% (small) respectively 49% (medium) of all SHI-financed WHPs were implemented in SME [2].”
  • Provide additional detail related to the theoretical framework, “the BIG model.” The section needs more review and references to the literature. Also, the categories need to be consistent. First it says the project is divided into five phases based on BIG model: A) finding, B) preparation, C) cooperative planning process, D) intervention process, and E) ensuring sustainability (Page 3). However, these phases do not line up with the later mention of 1) conception phase, 2) implementation phase and 3) evaluation phase (also on Page 3). This section needs more literature review, references, and consistency in description.
  • The five types of stakeholders needs clarification; company reps, exercise providers etc. List them out clearly.
  • On Page 4, there is a reference to “second node type;” however, I did not see a mention of the “first node type.” This needs to be clarified. “As a methodological tool, events (e.g. meetings and workshops) were defined as a SECOND NODE TYPE during the course of the project (see 2.3; table 1). The edges in our SNA represent participation of the corresponding actors in networking events.”
  • Please provide an organizational structure of the KomRueBer project at the outset. Who is responsible for sending the letter of intent and organizing the various events for this group of 90 different sized companies?
  • The five events listed in Table 1 need to be clearly described after the table.
  • If the Letter of Intent was sent to EVERYONE (all companies) and that was the first event, would that not have the highest participation? If so, would it not appear that participation decreased over time? This needs to be clarified.
  • Sections 2.2 and 2.3 pertaining to the SNA approach and SNA Analysis need additional SUBHEADINGS to better illustrate the methodology used for data collection and for data analysis in this study.
  • Use of two-mode network analysis is appropriate; the projection to one-mode network requires clarification. Again, the use of additional SUBHEADINGS to provide clarification can make it easier for people from all backgrounds to understand the social network analysis process.
  • In regard to results, can additional network results be provided by organizational SIZE and organizational TYPE? This would be helpful in understanding network evolution over time.
  • I would recommend revising the section heading from First Results to Results, since there is no second results section.
  • The Discussion section needs to include additional subsections on “summary of findings,” study limitations,” “implications for practice (companies);” and “implications for future research.”

Reviewer 3 Report

Researchers present a study on social network analysis applied to inter-firm network support for workplace health promotion. This research is part of a project known as the KoomRueBer Project.

In the manuscript they state that the main objective is to analyse the support of these inter-enterprise networks using Social Network Analysis (SNA).

The manuscript is very well structured, but I consider that there are some weaknesses that need to be improved.

1. In the introduction to sign that there are few SNA studies in the scope of their research, however, the citation used is from 2017. I advise you to look for more current citations and to include some of the studies conducted between 2017 and 2021 within your topic. This will not be detrimental to your own research, but it will help to have a more current and correct state of the art.

2. In the data collection, you do not indicate whether you did it through online questionnaires, paper questionnaires or how you did it.

3. The images are very improvable. Moreover, they look like they have been printed on paper and scanned.

4. Regarding the rest, I think the discussion is fine, but they should better justify, based on the most current studies, what innovation they offer in their study.

Round 2

Reviewer 2 Report

Thanks for your responsiveness to the comments.

Reviewer 3 Report

I thank the authors for the modifications made to the manuscript and for their work.

I consider that the article is now of publishable quality.